# Dose Reduction in Coronary Artery Calcium Scoring Using Mono-Energetic Images from Reduced Tube Voltage Dual-Source Photon-Counting CT Data: A Dynamic Phantom Study

**DOI:** 10.3390/diagnostics11122192

**Published:** 2021-11-25

**Authors:** Niels R. van der Werf, Margo van Gent, Ronald Booij, Daniel Bos, Aad van der Lugt, Ricardo P. J. Budde, Marcel J. W. Greuter, Marcel van Straten

**Affiliations:** 1Department of Radiology & Nuclear Medicine, Erasmus University Medical Center, 3015 GD Rotterdam, The Netherlands; r.booij@erasmusmc.nl (R.B.); d.bos@erasmusmc.nl (D.B.); a.vanderlugt@erasmusmc.nl (A.v.d.L.); r.budde@erasmusmc.nl (R.P.J.B.); marcel.vanstraten@erasmusmc.nl (M.v.S.); 2Department of Radiology, University Medical Center Groningen, University of Groningen, 9713 GZ Groningen, The Netherlands; m.van.gent01@umcg.nl (M.v.G.); m.j.w.greuter@umcg.nl (M.J.W.G.); 3Department of Robotics and Mechatronics, University of Twente, 7522 NB Enschede, The Netherlands

**Keywords:** X-ray computed tomography, calcium, coronary vessels, imaging phantoms, photon counting detector, radiation dose, image quality

## Abstract

In order to assess coronary artery calcium (CAC) quantification reproducibility for photon-counting computed tomography (PCCT) at reduced tube potential, an anthropomorphic thorax phantom with low-, medium-, and high-density CAC inserts was scanned with PCCT (NAEOTOM Alpha, Siemens Healthineers) at two heart rates: 0 and 60–75 beats per minute (bpm). Five imaging protocols were used: 120 kVp standard dose (IQ level 16, reference), 90 kVp at standard (IQ level 16), 75% and 45% dose and tin-filtered 100 kVp at standard dose (IQ level 16). Each scan was repeated five times. Images were reconstructed using monoE reconstruction at 70 keV. For each heart rate, CAC values, quantified as Agatston scores, were compared with the reference, whereby deviations >10% were deemed clinically relevant. Reference protocol radiation dose (as volumetric CT dose index) was 4.06 mGy. Radiation dose was reduced by 27%, 44%, 67%, and 46% for the 90 kVp standard dose, 90 kVp 75% dose, 90 kVp 45% dose, and Sn100 standard dose protocol, respectively. For the low-density CAC, all reduced tube current protocols resulted in clinically relevant differences with the reference. For the medium- and high-density CAC, the implemented 90 kVp protocols and heart rates revealed no clinically relevant differences in Agatston score based on 95% confidence intervals. In conclusion, PCCT allows for reproducible Agatston scores at a reduced tube voltage of 90 kVp with radiation dose reductions up to 67% for medium- and high-density CAC.

## 1. Introduction

Cardiovascular disease (CVD) is the most common cause of death in both the United States of America and Europe [1]. Among CVDs, ischemic heart disease as a consequence of intracoronary atherosclerosis remains the largest cause of death [2,3]. Computed tomography (CT) is the modality of choice to detect and quantify coronary artery calcium (CAC). Clinically, the Agatston score is used for patient risk stratification, with a strong association with future adverse cardiovascular events [4,5,6,7,8]. As a result, CAC assessment is recommended by several guidelines to improve clinical risk prediction in appropriately selected asymptomatic individuals, which therefore results in a high number of examinations [6,7,9].

Several studies have assessed potential radiation dose reduction techniques for CAC assessment with CT [10]. One technique is related to changes in tube potential. By reducing the overall energy or the spectrum of the x-ray beam, patient radiation dose can be reduced. As tissue attenuation, and corresponding CT numbers in Hounsfield units (HU) are energy dependent, modifications to the Agatston score methodology, such as HU-threshold optimization, may be required for reproducible results [11,12,13,14].

However, Agatston score methodology modifications might not be necessary for changes in tube potential for a new CT technology: photon-counting CT (PCCT) [15,16,17,18,19,20,21,22]. With PCCT, incoming photons are counted within predefined energy bins. These spectral data are subsequently used for image reconstruction of CT images for virtual mono-energetic (monoE) x-ray sources.

For one vendor, the clinical PCCT CAC protocol uses the standard acquisition tube potential of 120 kVp, with a subsequent monoE image reconstruction at 70 keV. The impact of reduced or spectral shaped (by means of tin filtration) tube potentials on CAC assessments with this new CT system remains unknown.

The aim of the current study is, therefore, to assess Agatston score reproducibility for reduced radiation dose CAC protocols with tube potential adjustments.

## 2. Materials and Methods

A routine clinical CAC protocol was used to scan an anthropomorphic thorax phantom (QRM-Thorax, QRM GmbH, Möhrendorf, Germany) on a first generation dual-source PCCT (NAEOTOM Alpha, Siemens Healthineers, Erlangen, Germany) (Table 1). A fat tissue equivalent extension ring was used to increase phantom dimensions to resemble a large patient size [23]. At the center of the thorax phantom, an artificial hydroxyapatite (HA) containing coronary artery was placed inside a water compartment. In total, three cylindrical calcifications of equal dimensions (5 mm diameter, 1 mm length) but different densities were used. The HA densities were 196 ± 3, 408 ± 2, and 800 ± 2 mg/cm^3^, or low, medium, and high density, respectively. Besides a static scan, a dynamic scan was made, whereby the artery was translated at 20 mm/s by a computer-controlled lever (QRM-Sim2D, QRM GmbH, Möhrendorf, Germany). The static and dynamic scan corresponded to 0 and 60–75 beats per minute (bpm), respectively [24,25]. The ECG trigger from the ECG-output of the computer-controlled lever was used to ensure data acquisition only during linear motion of the artificial coronary artery [25].

Following the reference CAC protocol scan at 120 kVp, additional acquisitions at reduced tube potentials of 90 and Sn100 kVp were made. For all tube potentials, tube current modulation with image quality level 16 (CareIQ, Siemens Healthineers, Erlangen, Germany) was used as the standard dose. In addition, radiation dose in terms of the volumetric CT dose index (CTDIvol) was further reduced for the 90 kVp acquisition to 75% and 45% of the standard dose, with tube current reduction. Overall, this resulted in five combinations of tube potential and tube current: (1) reference (120 kVp, 20 mAs), (2) 90 kVp standard dose, (3) 90 kVp 75% dose, (4) 90 kVp 45% dose, and (5) Sn100 kVp standard dose. For each combination, scans were repeated five times, with manual repositioning between each scan (2 mm translation, 2 degrees rotation). Each scan was reconstructed at a monoE level of 70 keV.

For each reconstruction, Agatston scores were automatically determined with the use of a previously validated, open-source Python script (Python version 3.7) [26]. A threshold of 130 HU was used to discriminate CAC from the background signal.

In addition to the Agatston score, several image quality metrics were calculated. First of all, total image noise (standard deviation (SD)) was calculated from a large uniform region of interest (ROI) (128 × 128 voxels). For this same ROI, a so-called background Agatston score (BAS) was determined [26,27]. The BAS was calculated by summation of the number of voxels that exceed the CAC threshold of 130 HU within this ROI. For reconstructions with a non-zero BAS, noise levels were too high, which led to false positive CAC assessment. Third, noise-power-spectra (NPS) were determined in the same slice. Finally, contrast-to-noise ratios (CNRs) were determined for the three CAC densities, by dividing the absolute difference in HU between CAC and background material by the total image noise (SD).

For each heart rate, Agatston scores for the reduced kVp and reduced dose were compared to the reference at 120 kVp with standard dose. We calculated the 95% confidence interval for the mean difference. If there were deviations in Agatston score < 10%, then we deemed these differences to be not clinically relevant.

## 3. Results

### 3.1. Radiation Dose

Automatic exposure control at standard dose (IQ level 16) with tube voltages 120 kVp (reference scan), 90 kVp, and Sn100 kVp resulted in a radiation dose of 4.06 mGy, 2.97 mGy, and 2.21 mGy, respectively (Table 1). Accordingly, scans acquired at 90 kVp with 75% and 45% of the standard dose resulted in a radiation dose of 2.26 mGy and 1.33 mGy, respectively.

### 3.2. Image Quality

Mean and standard deviation (SD) image noise levels were 25.3 ± 0.3, 25.0 ± 2.0, 27.0 ± 2.3, 30.7 ± 3.0, and 25.7 ± 1.9 for the reference (120 kVp standard dose), 90 kVp standard dose, 90 kVp 75% dose, 90 kVp 45% dose, and Sn100 kVp standard dose, respectively. These noise levels only resulted in BAS > 0 for 2 out of 100 scans, namely for the 90 kVp at 45% radiation dose.

Example images and NPS for all combinations of tube potential, radiation dose, and heart rate are shown in Figure 1. Overall, noise spatial frequency decreased with decreasing tube potential.

With comparable image noise for the acquisitions, standard dose CAC CNRs were comparable for the 120, 90, and Sn100 kVp acquisition (Figure 2). With respect to the reference, CNR decreased with increasing image noise, for the reduced dose acquisitions at 90 kVp. At increased heart rate, CNR increased for Sn100 kVp acquisitions for all CAC densities.

### 3.3. Influence on Agatston Scores

For the static phantom, reference Agatston scores as mean and SD were 79.3 ± 4.4, 358.74 ± 9.68, and 438.26 ± 5.44 for the low-, medium-, and high-density CAC, respectively. Reference Agatston scores for the calcification at 60–75 bpm were 65.0 ± 4.1, 346.7 ± 12.6, and 465.5 ± 8.6 for the low-, medium-, and high-density CAC, respectively.

Relative differences of the reference Agatston scores with scores from the other protocols are shown in Figure 3. Firstly, for the low-density calcification, variability in Agatston score differences with the reference was large for all protocols.

Secondly, for the static medium- and high-density calcifications, Agatston score differences with the reference were not clinically relevant (<10%). At 60–75 bpm, similar results were shown, with no significantly different Agatston scores in comparison with the reference for all protocols.

## 4. Discussion

The main finding of this study is that for medium- and high-density CAC, monoE reconstructions at 70 keV for PCCT acquisitions at various tube potentials allow for reproducible Agatston scores. With this, patient radiation dose could be reduced up to 67%, without clinically relevant changes to the resulting CAC assessment for these CAC densities for 90 kVp acquisitions. However, variations in Agatston score differences with the reference for low-density CAC were large. In comparison with the reference at 120 kVp, CNR did not decrease for 90 and Sn100 kVp. Overall, noise spatial frequency increased with increasing tube potential.

To the best of our knowledge, the current study is the first to assess Agatston score reproducibility for monoE reconstructions acquired with various tube potentials. The main reason for this is that conventional dual-source CT sacrifices temporal resolution when acquiring spectral data. As high temporal resolution is essential for reproducible CAC quantification, spectral mode is not recommended for CAC studies [25,28,29]. For PCCT, however, spectral data is available at high-temporal resolution as well.

While reproducible Agatston scores are key for robust risk stratification, many factors have previously been shown to influence Agatston score reproducibility. These factors include patient size, heart rate, CT system, scan starting position, slice thickness, and CAC quantification parameters [25,26,29,30,31,32,33,34,35]. These parameters especially influence low-density CAC Agatston scores, for which the CT numbers just exceed the CAC scoring threshold of 130 HU. In our current study, large variations in Agatston scores for the low-density CAC resulted in clinically relevant differences in Agatston scores for all reduced tube potential protocols. A post-hoc power analysis revealed that particularly in this low-density category we were underpowered (1–β: 0.31) to detect ‘clinically’ meaningful differences. For low-density CAC, the threshold of deviations in Agatston score > 10% to indicate clinical relevance may be too strict, especially considering the large number of parameters that influence this measurement.

Many previous studies have evaluated the potential of CAC assessment at reduced radiation dose [10]. Despite using adjusted CAC scoring thresholds for reduced tube potential acquisitions, both Thomas et al. and Marwan et al. found overestimations in Agatston scores for phantom and patients [13,36]. Contrarily, Gräni et al. showed that Agatston scores were underestimated with adjusted CAC scoring thresholds, in comparison with the standard 120 kVp protocol [11]. The dose reduction potential (57–65%) of these studies was comparable to our results. For our study, conventional CAC scoring thresholds of 130 HU could be used for all acquisitions, as all scans were reconstructed at a monoE level of 70 keV. For the same PCCT system, Eberhard et al. showed a different approach for radiation dose reduction, with different monoE levels and the use of iterative reconstruction [37]. For our study, the clinical CAC protocol (FBP and 70 keV) was used. A combination of other monoE levels, iterative reconstruction, and tube voltage reduction acquisitions may result in further radiation dose reductions.

This study has limitations that merit consideration. First, an anthropomorphic phantom with artificial CAC containing coronary arteries and artificial tissue-simulating materials was used for the current study. The densities of the artificial CAC were mixtures of HA and so-called solid water. The mass of the calcifications was in the range that is observed in patients [38]. Second, the complex in vivo motion of coronary arteries was simulated by the phantom as a translation of the coronary artery in one direction, perpendicular to the scan plane. The constant linear motion of our phantom was deemed sufficient, as the scan times were relatively short as a result of fast gantry rotation times [24]. Third, our CAC contrast calculation was based on all voxels that exceeded the CAC threshold in multiple slices. The reason for this approach is the small number of voxels for each slice that exceed the CAC scoring threshold, due to the small diameter of the calcification. Consequently, the resulting CNR was underestimated compared to what would be expected for the known CAC densities. Fourth, only discrete radiation dose reduction steps were used for the current study. For the 90 kVp acquisitions, radiation dose reduction was at 75% and 45% for the standard dose. The latter resulted in BAS > 0 for one repetition. Optimal radiation dose reduction might therefore lie between 45% and 75% of the standard radiation dose for 90 kVp acquisitions. Fifth, this study was underpowered to draw conclusions on low-density calcifications. Extensive follow-up studies are needed to validate our results in vivo. Sixth, only one PCCT system was used for our current study. However, this PCCT system is currently the only clinically available PCCT system that can provide monoE reconstructions at high-temporal resolution.

## 5. Conclusions

In conclusion, PCCT allows for reproducible Agatston scores at radiation dose reductions up to 67% for moving calcifications of medium and high density, when using a reduced tube potential acquisition of 90 kVp, reconstructed at a monoE level of 70 keV.

## Figures and Tables

**Figure 1 diagnostics-11-02192-f001:**
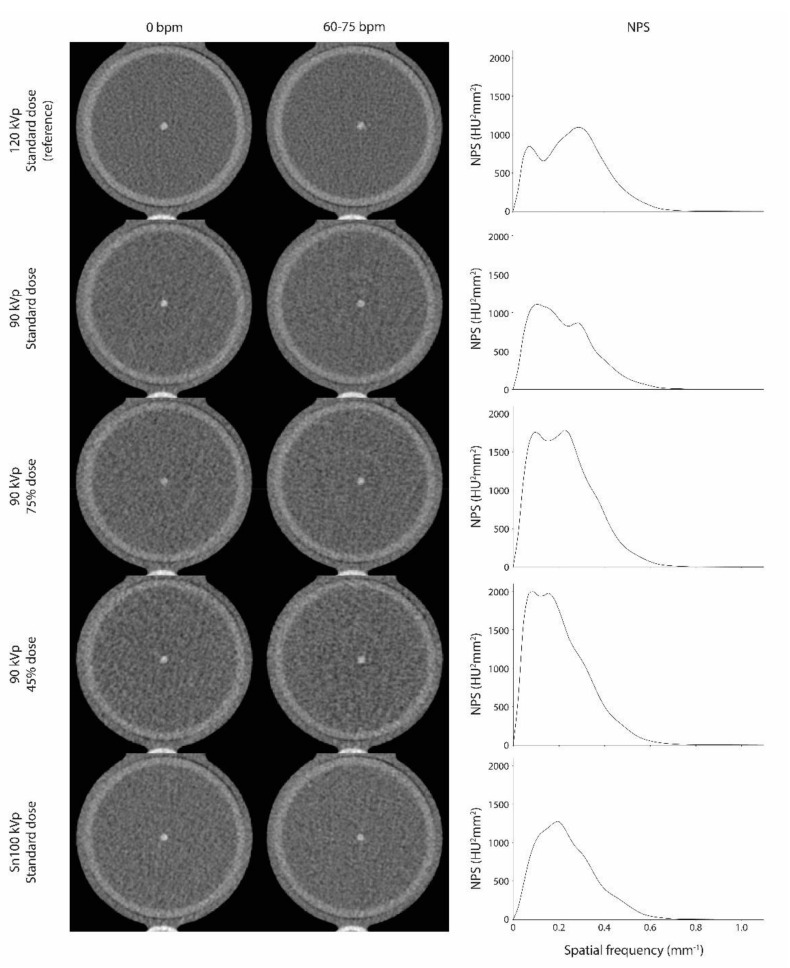
MonoE reconstructed images of the low-density CAC at 70 keV for all combinations used of tube potential, radiation dose, and heart rate. Noise-power-spectra (NPS) are indicated in the right column.

**Figure 2 diagnostics-11-02192-f002:**
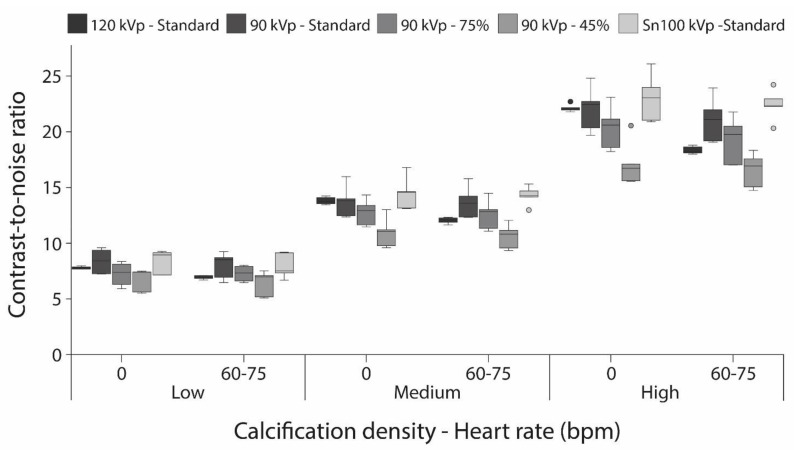
Contrast-to-noise ratio box and whisker plots for the low-, medium-, and high-density calcification, translated at 0 and 60–75 bpm for all combinations of tube potential and radiation dose.

**Figure 3 diagnostics-11-02192-f003:**
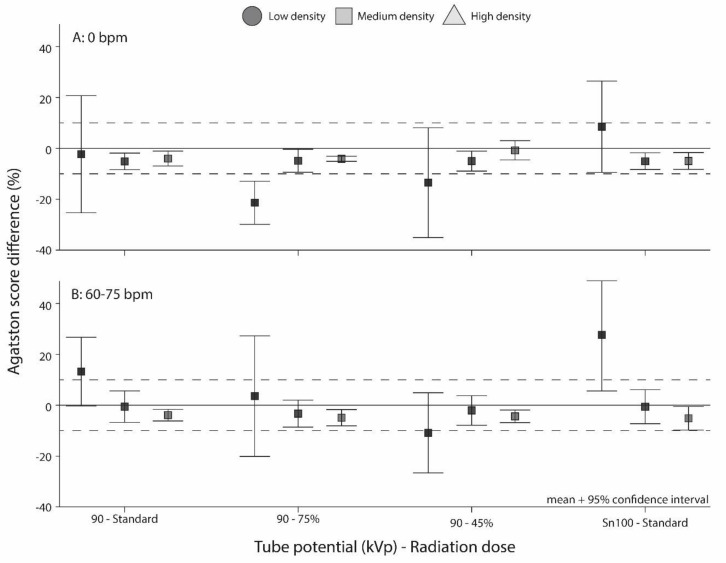
Mean and 95% confidence interval Agatston score deviation in % from the reference (120 kVp IQ level 16) with (**A**) stationary calcifications and (**B**) moving calcifications at 60–75 bpm. The dashed lines indicate deviations of ±10%, which is the threshold for clinically relevant differences.

**Table 1 diagnostics-11-02192-t001:** Acquisition and reconstruction parameters.

Parameter	Reference Scan	90 kVp Standard Dose	90 kVp 75% Dose	90 kVp 45% Dose	Sn100 kVp Standard Dose
CT system	PCCT	PCCT	PCCT	PCCT	PCCT
Technique	Axial	Axial	Axial	Axial	Axial
Tube voltage (kVp)	120	90	90	90	100 + Sn filter
Effective tube current time product (mAs)	20 ^1^	45 ^1^	34	20	134 ^1^
Automatic exposure control	Off	Off	Off	Off	Off
Collimation (mm)	144 × 0.4	144 × 0.4	144 × 0.4	144 × 0.4	144 × 0.4
Field of view (mm)	220	220	220	220	220
Rotation time (s)	0.25	0.25	0.25	0.25	0.25
Slice thickness/increment (mm)	3.0/1.5	3.0/1.5	3.0/1.5	3.0/1.5	3.0/1.5
Reconstruction kernel	Qr36	Qr36	Qr36	Qr36	Qr36
Matrix size (pixels)	512 × 512	512 × 512	512 × 512	512 × 512	512 × 512
Reconstruction	FBP ^2^	FBP ^2^	FBP ^2^	FBP ^2^	FBP ^2^
monoE level (keV)	70	70	70	70	70
Repetitions	5	5	5	5	5
Phantom speed	0 & 20 mm/s	0 & 20 mm/s	0 & 20 mm/s	0 & 20 mm/s	0 & 20 mm/s
CTDI_vol_ (mGy)	4.06	2.97	2.26	1.33	2.21

^1^ Based on the vendor recommended reference CareIQ level 16. ^2^ FBP: filtered back projection. The setting used was actually Quantum Iterative Reconstruction (QIR, Siemens Healthineers) off, which is comparable to a conventional reconstruction in terms of the expected noise level.

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
