# Peer review of "Dose Reduction in Coronary Artery Calcium Scoring Using Mono-Energetic Images from Reduced Tube Voltage Dual-Source Photon-Counting CT Data: A Dynamic Phantom Study"

_diagnostics, 2021, doi:10.3390/diagnostics11122192_

Round 1

Reviewer 1 Report

This phantom study adressed the potential of radiation dose reduction for CT calcium scoring with reduced tube voltage on a novel dual-source photon-counting CT. It is well written and highly relevant but I recommend to adress a few points.

Title: ok

Abstract/Conclusion: « Radation dose reduction up to 67% for medium to high density calcifications ».  This conclusion may be true, but in the real world we also encounter patients with low density calcifications. What is the conclusion for CT calcium scoring at reduced tube voltages on a photon-counting CT in general ?

Introduction:

P2l53: Previously, it was reported that CAC sores from images at 70keV and FBP are comparable to CAC scores from images at 65keV and quantum iterative image recontruction at strength levels 3 or 4 (doi: 10.3390/diagnostics11091708). Please refer to this paper and state why you chose to assess only 70keV images. May application of 65keV images and iterative image reconstruction even allow for CT CAC scoring at lower radiation doses ? What do you think ? This point is also relevant for your discussion on page 7l180.

Methods:

As far as I understand this topic and although it has the same label, FBP on a photon-counting CT is not the same as FBP on an energy-integrating detector CT. Please address this issue in 1-2 sentences or briefly discuss this issue in the appropriate manuscript section.

P3l78 : Why did you choose an IQ level of 16 ?

P3l89-96 : Who performed measurements? Please be clear.

P3l92: Please define the background Agatston Score in 1-2 sentences.

P3l100 : It is well known that scanning with different CT systems result in different CAC scores. Therefore, it seems tob e ok to introduce a threshold of 10% to define relevant changes. However, in reference 28, this threshold was also arbitrarily chosen and not supported by evidence. Therfore, I suggest removing this reference (please also remove this reference in the discussion).

Results : Why didn’t you use an energy-integrating detector CT scanner for reference ?

Discussion: P6l160-p7l172: You discuss deviation of low density calcifications. What is your point for future studies adressing this issue ?

Author Response

Comment 1:

Radiation dose reduction up to 67% for medium to high density calcifications.  This conclusion may be true, but in the real world we also encounter patients with low density calcifications. What is the conclusion for CT calcium scoring at reduced tube voltages on a photon-counting CT in general?

Response 1:

We thank the reviewer for the opportunity to improve our manuscript. Our study, as indicated in the discussion section, was underpowered for low density calcifications. We calculated that 15 repetitions would be needed for a power of more than 80% to detect ‘clinically’ meaningful differences, if present. However, many factors influence reproducibility of Agatston scores for these calcifications. For low density CAC, the threshold of deviations in Agatston score > 10% to indicate clinical relevance may be too strict, especially considering the large number of parameters which influence this measurement. Overall, the current study should be seen as a feasibility study for dose reduction, which should be validated in-vivo for different dose reduction levels, and for all calcification densities. This was added as a limitation (page 8, lines 218-220): “Fifth, this study was underpowered to draw conclusions on low density calcifications. Extensive follow-up studies are needed to validate our results in-vivo.”

Comment 2:

Previously, it was reported that CAC sores from images at 70keV and FBP are comparable to CAC scores from images at 65keV and quantum iterative image reconstruction at strength levels 3 or 4 (doi: 10.3390/diagnostics11091708). Please refer to this paper and state why you chose to assess only 70keV images. May application of 65keV images and iterative image reconstruction even allow for CT CAC scoring at lower radiation doses ? What do you think ? This point is also relevant for your discussion on page 7l180

Response 2:

Thank you for this suggestion. We think that at 65 keV, calcium attenuation will increase and CT numbers will increase. Without changing the CAC threshold of 130HU, this will increase CAC sensitivity, and therefore low density CAC detection. However, applying iterative reconstruction, or changing monoE reconstruction levels should be performed carefully, due to its - calcification density specific – effect on Agatston scores. For our study, we used vendor recommended clinical protocols, which use a reconstruction of 70 keV for CAC assessment. We have added the reference to our manuscript (page 8, lines 197-201): “For the same PCCT system, Eberhard et al showed a different approach for radiation dose reduction, with different monoE levels and the use of iterative reconstruction [39]. For our study, the clinical CAC protocol (FBP and 70 keV) was used. A combination of other monoE levels, iterative reconstruction, and tube voltage reduction acquisitions may result in further radiation dose reductions.”

Comment 3:

As far as I understand this topic and although it has the same label, FBP on a photon-counting CT is not the same as FBP on an energy-integrating detector CT. Please address this issue in 1-2 sentences or briefly discuss this issue in the appropriate manuscript section.

Response 3:

We agree with the reviewer that FBP on PCCT is not the same as FBP on CT. We have clarified this in our manuscript (page 3, caption of table 1): “FBP = Filtered back projection. The used setting was actually Quantum Iterative Reconstruction (QIR, Siemens Healthineers) off, which is comparable to a conventional reconstruction in terms of the expected noise level”

Comment 4:

Why did you choose an IQ level of 16 ?

Response 4:

Thank you for this question. The IQ level of 16 is recommended by the vendor in the clinical CAC protocol. This information was added to the caption of table 1: “Based on the vendor recommended reference CareIQ level 16”

Comment 5:

Who performed measurements? Please be clear.

Response 5:

We thank the reviewer for this comment. As indicated in lines 91-93 of the methods section, all measurements were performed fully automatically with the use of a previously validated, in-house developed Python script described in doi:10.1002/mp.14912.

Comment 6:

Please define the background Agatston Score in 1-2 sentences.

Response 6:

Thank you for the opportunity to clarify the BAS. We have added the following line to our manuscript (page 3, lines 97-98): “The BAS was calculated by summation of the number of voxels which exceed the CAC threshold of 130 HU within this ROI.”

Comment 7:

It is well known that scanning with different CT systems result in different CAC scores. Therefore, it seems to be ok to introduce a threshold of 10% to define relevant changes. However, in reference 28, this threshold was also arbitrarily chosen and not supported by evidence. Therefore, I suggest removing this reference (please also remove this reference in the discussion).

Response 7:

We have removed the reference, as suggested.

Comment 8:

Why didn’t you use an energy-integrating detector CT scanner for reference ?

Response 8:

Thank you for this question. We did not use a conventional CT, as those systems do not allow for reconstructions with comparable CT numbers when altering tube potential, or without sacrificing temporal resolution. The only solution is to use different CAC thresholds, as shown by for example Gräni et al which was referred to by us in our discussion section.

Comment 9:

You discuss deviation of low density calcifications. What is your point for future studies addressing this issue ?

Response 9:

For our response on this comment, we would like to refer to our response on comment 1.

Reviewer 2 Report

The manuscript significantly improved after the revision. I have no further comments or edits.

Author Response

Thank you.

Reviewer 3 Report

The authors evaluated Agatston score reproducibility for
reduced radiation dose CAC protocols with tube potential adjustments.

They concluded that PCCT allowed for reproducible Agatston scores at a reduced tube voltage of 90kVp with radiation dose reductions up to 67% for medium and high density CAC.

I have the following concerns:

  1. How many procedures were performed?
  2. Was the study retrospective or prospective?
  3. Was the informed consent signed?
  4. Please include data on Ethics.
  5. What were the practical implications of the study?

Author Response

Comment 1:

How many procedures were performed?

Response 1:

Thank you for this question. In total, 50 scans (5 tube potential / dose combinations, 5 repetitions, 2 heart rates) were performed.

Comment 2:

Was the study retrospective or prospective?

Response 2:

This study was a phantom study, and therefore prospective.

Comment 3:

Was the informed consent signed?

Response 3:

As we have used only phantom data for the current study, no informed consent was necessary.

Comment 4:

Please include data on Ethics.

Response 4:

For our response on this comment, we would like to refer to our response on comment 3 (from reviewer #2).

Comment 5:

What were the practical implications of the study?

Response 5:

Thank you for this question. This study should be seen as a first step towards clinical radiation dose reduction for patients. Before this can be done, extensive follow-up studies on patients have to be performed, as indicated in our fifth limitation of our manuscript.

Round 2

Reviewer 1 Report

Thanks for the revision

Reviewer 3 Report

All my comments have been adequately addressed. I have no further comments.